# The *rpoS* gene confers resistance to low osmolarity conditions in *Salmonella enterica* serovar Typhi

Eamon Gibbons[1]☯, Mehbooba Tamanna[1,2]☯, Bobby J. Cherayil [1,3]*

1 Mucosal Immunology and Biology Research Center, Massachusetts General Hospital, Charlestown, Massachusetts, United States of America, 2 Medical Sciences Program, Boston University School of Medicine, Boston, Massachusetts, United States of America, 3 Department of Pediatrics, Harvard Medical School, Boston, Massachusetts, United States of America

☯ These authors contributed equally to this work.
* cherayil@helix.mgh.harvard.edu

**Data Availability Statement:** All relevant data are within the paper and its Supporting information files.

## Abstract

*Salmonella enterica* serovars Typhimurium and Typhi are enteropathogens that differ in host range and the diseases that they cause. We found that exposure to a combination of hypotonicity and the detergent Triton X-100 significantly reduced the viability of the *S.* Typhi strain Ty2 but had no effect on the *S.* Typhimurium strain SL1344. Further analysis revealed that hypotonicity was the critical factor: incubation in distilled water alone was sufficient to kill Ty2, while the addition of sodium chloride inhibited killing in a dose-dependent manner. Ty2's loss of viability in water was modified by culture conditions: bacteria grown in well-aerated shaking cultures were more susceptible than bacteria grown under less aerated static conditions. Ty2, like many *S.* Typhi clinical isolates, has an inactivating mutation in the *rpoS* gene, a transcriptional regulator of stress responses, whereas most *S.* Typhimurium strains, including SL1344, have the wild-type gene. Transformation of Ty2 with a plasmid expressing wild-type *rpoS*, but not the empty vector, significantly increased survival in distilled water. Moreover, an *S.* Typhi strain with wild-type *rpoS* had unimpaired survival in water. Inactivation of the wild-type gene in this strain significantly reduced survival, while replacement with an arabinose-inducible allele of *rpoS* restored viability in water under inducing conditions. Our observations on *rpoS*-dependent differences in susceptibility to hypotonic conditions may be relevant to the ability of *S.* Typhi and *S.* Typhimurium to tolerate the various environments they encounter during the infectious cycle. They also have implications for the handling of these organisms during experimental manipulations.

## Introduction

*Salmonella enterica* serovar Typhi (*S.* Typhi) and *Salmonella enterica* serovar Typhimurium (*S.* Typhimurium) are closely related Gram-negative bacteria that differ in host range and the diseases that they cause. Both organisms are enteropathogens that are transmitted through contaminated food or water. However, *S.* Typhi infects only humans and causes typhoid fever, a

**Funding:** BJC was supported by grant numbers R01AI089700 and R21AI155593 from the National Institute of Allergy and Infectious Diseases (https://www.niaid.nih.gov). The funder had no role in study design, data collection and analysis, decision to publish, or preparation of the manuscript.

**Competing interests:** The authors have declared that no competing interests exist.

systemic illness involving spread of the pathogen to extra-intestinal tissues. The infection is potentially fatal if it is not treated appropriately. Accordingly, typhoid is responsible for significant morbidity and mortality in the developing world, mainly in young children, and has become particularly problematic because of the emergence of multidrug resistance [1, 2]. *S.* Typhimurium infects a broad range of hosts globally, including humans, farm animals and laboratory mice [3–6]. In humans, *S.* Typhimurium infection is usually confined to the intestine, causing a robust local inflammatory response that manifests as acute, self-limiting gastroenteritis [3]. It rarely spreads systemically and generally does not require specific treatment, although certain pathovars of *S.* Typhimurium can cause an invasive, typhoid-like systemic disease in children and immunocompromised adults in sub-Saharan Africa [7]. The mechanisms responsible for these differences between *S.* Typhi and *S.* Typhimurium are not completely understood.

During infection of their hosts, enteropathogens like *S.* Typhi and *S.* Typhimurium have to adapt to changing environments as they pass through the gastrointestinal tract and then interact with epithelial and immune cells [8–12]. The challenges that they have to contend with include alterations in temperature, pH, osmolarity and oxygen and nutrient concentrations, as well as potentially toxic molecules such as bile salts, antimicrobial peptides and oxidative radicals [10, 11, 13–19]. In addition, they have to survive, at least for some time, in the external environment in order to pass from one host to another. This requirement may be particularly relevant to *S.* Typhi, which does not have a non-human animal host that can act as an intermediary in transmitting infection. Although direct fecal-oral transmission of *S.* Typhi between humans may occur, indirect transmission via consumption of untreated water from wells, ponds and rivers probably plays an even more important role [20–23]. Thus, the pathogen must have the ability to adjust to and tolerate the low osmolarity of such freshwater sources. Indeed, it has been shown that *S.* Typhi adapts to water by modulating the expression of a large number of genes, including some that are essential for viability under such conditions [24].

One of the genes that plays a key role in adapting to multiple environmental stresses in *Salmonella* is *rpoS*, which encodes the stationary phase alternate sigma factor RpoS [25]. Disruption of the *rpoS* gene in *S.* Typhimurium has been shown to impair survival in the presence of low pH, nutrient deprivation and oxidative agents, and to attenuate virulence in mice [26, 27]. Studies with *S.* Typhimurium strains carrying a mutant allele of *rpoS* have led to similar findings, along with demonstration of significantly reduced colonization of gut-associated lymphoid tissues, spleen and liver by the mutant bacteria despite their normal invasiveness and ability to survive in macrophages [28, 29]. These effects have been linked, either experimentally or presumptively, to the involvement of *rpoS* in controlling the expression of a large number of genes that contribute to metabolic processes, resistance to oxidative molecules, DNA damage repair, membrane transport, and various aspects of virulence [30–33]. The *rpoS* gene has also been shown to play important roles in *S.* Typhi, including in resistance to acid, starvation and reactive oxygen and nitrogen species, synthesis of the Vi capsular polysaccharide, expression of the HlyE hemolysin and the OsmY periplasmic protein, macrophage cytotoxicity and adhesion to epithelial cells [34–39]. In addition, the wild-type or mutant *rpoS* status of *S.* Typhi vaccine strains has been shown to affect their ability to survive stresses associated with the host environment, and to influence the kinds of immune responses that they elicit in mice [40–43].

Adding to the above observations, the experiments described in the present work show that the *S.* Typhi *rpoS* gene is involved in the ability of the pathogen to survive under low osmolarity conditions. The results are relevant to our understanding of how *S.* Typhi deals with the transition to the external environment during host-to-host transmission, and also have implications for handling of the organism during experimental manipulations.

## Materials and methods

### Bacterial strains

The wild-type *S.* Typhi strain Ty2 was obtained from the American Type Culture Collection, Manassas, VA, while the wild-type *S.* Typhimurium strain SL1344 was originally provided by Dr. Beth McCormick, University of Massachusetts Medical Center, Worcester, MA. The *S.* Typhi ISP1820 strains χ3744 (wild-type *rpoS* gene), χ9061 (*rpoS* inactivated by phage-mediated introduction of a mutant allele expressing a truncated protein) and χ9066 (arabinose-inducible wild-type *rpoS* generated by allelic exchange) were constructed in the laboratory of Dr. Roy Curtiss, III as detailed earlier [36], and were kindly provided by Drs. Curtiss and Soo-Young Wanda. Unless indicated otherwise, bacteria were routinely grown at 37˚C in Lysogeny Broth (LB, Miller formulation, 10 g/l NaCl, Sigma Chemical Co., St. Louis, MO) in sterile 12 ml polypropylene tubes (Fisher Scientific, Manassas, VA), using 2 ml of LB in loosely capped tubes or 10 ml of LB in tightly capped tubes for shaking or static cultures, respectively [44]. The cultures were shaken at 220 rpm when necessary. Ampicillin was added to a concentration of 100 μg/ml when plasmid selection was required. As previously described, arabinose was added to a concentration of 0.2% to induce *rpoS* expression in χ9066 [36].

### Bacterial viability assays

Bacterial cultures were grown overnight for 18–20 hours at 37˚C in LB under static or shaking conditions as specified in individual experiments. Aliquots of each culture were centrifuged, and the pelleted bacteria were washed once with phosphate buffered saline (PBS). After an additional centrifugation, the washed bacteria were resuspended in PBS at a density of approximately $5 \times 10^8$ colony forming units per ml (cfu/ml), as determined by preliminary titering experiments. Ten μl aliquots of the bacterial suspensions were added to replicate sterile microcentrifuge tubes containing 200 μl of either PBS or the test solution indicated in each experiment and mixed well. The tubes were incubated on ice or in a 37˚C water bath for 1 hour or for different times indicated in specific experiments. At the end of the incubation period, serial dilutions of the incubation mixes were made in PBS and plated on LB agar to determine the number of surviving bacteria. The number of viable bacteria in the test condition was expressed as a percentage of the number in PBS. In experiments involving plasmid-transformed bacteria, ampicillin at 100 μg/ml was included in culture media and buffers to maintain selection for the plasmid at all steps.

### Bacterial growth curves

Overnight shaking cultures of the various strains grown in LB were diluted 1000-fold into triplicate 1.2 ml aliquots of fresh medium distributed in the wells of 24-well tissue culture plates. The plates were incubated at 37˚C with shaking at 220 rpm. To generate growth curves, 100 μl of the culture from each well was removed at hourly intervals and transferred to a 96-well microplate for measurement of absorbance at 595 nm in a Spectramax ABS plate reader (Molecular Devices, San Jose, CA). To ensure that the culture volume was not depleted, separate plates were used for the measurements at 0–6 hours, 7–16 hours and 24–26 hours. The effect of low osmolarity was determined by comparing growth in standard LB (10 g/l of NaCl, osmolarity approximately 400 mosmol/l) versus hypotonic LB (no NaCl, osmolarity approximately 100 mosmol/l) [45, 46].

### Gene expression analysis

Five hundred μl of overnight shaking cultures of the relevant bacterial strains grown in standard LB were used to prepare RNA with the RNAprotect Bacteria Reagent and RNeasy Plus

Mini Kit (Qiagen, Germantown, MD) according to the manufacturer's instructions. Approximately 500 ng of the RNA samples were treated with RQ1 RNAse-free DNAse (Promega, Madison, WI) for 1 hour at 37˚C to eliminate any contaminating genomic DNA. The DNAse-treated RNA was then reverse transcribed using the iScript cDNA synthesis kit (Bio-Rad, Hercules, CA) according to the manufacturer's directions, and the cDNA subjected to real-time quantitative PCR amplification with the iQ SYBR Green Supermix kit (Bio-Rad, Hercules, CA) in an Applied Biosystems QuantStudio 3 (Fisher Scientific, Manassas, VA). The primers used for amplification were designed using Primer-BLAST (National Center for Biotechnology Information, National Library of Medicine, Bethesda, MD) and were specific for the *Salmonella enterica* genes *katE* (forward, 5' `CGTACGGTATCAGCAGACCC` 3', reverse, 5' `CCCGAGACCGCGAATGATAA` 3') and *otsA* (forward, 5' `TTCGAGT CAAACGCGAGTCA` 3', reverse, 5' `ACGTAGGCGCAATTTGGGTA` 3'), and for pan-bacterial 16S rRNA (forward, 5' `TCCTACGGGAGGCAGCAGT` 3', reverse, 5' `GGACTACAGGG TATCTAATCCTGTT` 3'). Forty cycles of amplification were carried out as previously described, with denaturation at 95˚C for 30 seconds, annealing at 55˚C for 30 seconds and extension at 72˚C for 1 minute [44]. The specificity of the amplification was confirmed by melting curve analysis: all products yielded single, sharp melting curve peaks. Expression was determined using the $2^{-\Delta\Delta Ct}$ method, with the levels of the transcript of interest normalized to 16S rRNA and calculated relative to the levels in SL1344 or Ty2 as indicated in the specific experiment.

## Cloning of the *S.* Typhimurium *rpoS* gene

Genomic DNA was prepared from an overnight culture of SL1344 using the UltraClean Microbial DNA kit (Qiagen, Germantown, MD). It was used as the template to PCR amplify the entire coding region of the *rpoS* gene (gene ID 11768183) with primers that incorporated EcoR1 and Not1 restriction sites at the 5' and 3' ends, respectively (forward, 5' `GTAGGGAATTCGGGTAGGAGCCACCTTATGAG` 3', reverse, 5' `ATAGCGGCCGCTACT TACTCGCGGAACAGCGC` 3'). The approximately 1 kb amplification product was digested with EcoR1 and Not1, gel purified and ligated to the similarly digested, ampicillin resistance-encoding bacterial expression vector pBH [47]. pBH is a high-copy plasmid (Roche, Nutley, NJ) in which expression of the cloned gene is under the control of a *trp-lac* promoter. Since Ty2 does not have the *lac* repressor, pBH-mediated expression is constitutive. We have used the vector previously to obtain robust and reliable expression of multiple genes in *Salmonella* [47]. The ligation mix was transformed into competent *E. coli* DH5α (Fisher Scientific, Manassas, VA) and the transformants were screened for the presence of plasmid containing the *rpoS* insert. One such plasmid (designated pBHSTm*rpoS*) was prepared in larger scale using a midiprep column (Qiagen, Germantown, MD) and sequenced to confirm the identity of the insert and to verify the absence of any mutations. pBHSTm*rpoS*, as well as the empty vector pBH, were transformed separately into Ty2 by electroporation as described previously to generate the transformants Ty2/pBHSTm*rpoS* and Ty2/pBH [44].

## Statistical analysis

Experiments to assess bacterial viability, growth and gene expression were carried out with 2 to 4 biological replicates, and the data from multiple experiments pooled for analysis. The scatter histogram plots in the figures show means +/- standard deviations with each symbol corresponding to an individual replicate. Statistical analysis was carried out with Prism 6.0 software (GraphPad Software, Inc., San Diego, CA). One-way ANOVA with correction for multiple comparisons was used to compare more than 2 experimental groups. The student's t test was

used when only 2 groups were involved. In both cases, p < 0.05 was considered significant. The actual p values are indicated in the legends of individual figures, along with the number of replicates.

## Results

### Viability of S. Typhi strain Ty2 is compromised by Triton X-100 and hypotonic conditions

Triton X-100 is a non-ionic detergent that is widely used at different concentrations (usually 0.5–1%) to lyse mammalian cells in gentamicin protection assays for the assessment of *Salmonella* invasiveness [48]. In initial experiments, we tested the ability of *S.* Typhimurium and *S.* Typhi prepared from overnight static cultures to survive incubation in 1% Triton X-100. The viability of the *S.* Typhimurium strain SL1344 was not affected by exposure to this detergent: equivalent numbers of surviving bacteria were recovered following a 1 hour incubation on ice in either 1% Triton X-100 in water or in PBS (Fig 1A). However, the *S.* Typhi strain Ty2 behaved very differently under these conditions: less than 5% of the bacteria survived the incubation in Triton/water relative to incubation in PBS (Fig 1A). This effect was seen with Triton concentrations as low as 0.1% and with incubation times as short as 15 minutes (Fig 1B). Moreover, similar results were obtained when we used Ty2 bacteria prepared from well-aerated overnight shaking cultures (Fig 1C), and when the incubation in Triton/water was carried out at 37°C (Fig 1D).

The compromised viability of Ty2 seen in these experiments could be the result of the detergent itself, the low osmolarity of the detergent solution (which was made in distilled water), or a combination of the two. To discriminate between these possibilities, we examined the ability of the bacteria (prepared from overnight static cultures) to survive a 1 hour incubation on ice in 1% Triton X-100 dissolved in water or PBS, or in plain sterile distilled water, relative to incubation in PBS. In contrast to the effects of incubation in Triton/water, we found that incubation in Triton/PBS had no effect on Ty2 viability, while incubation in just distilled water reduced viability by about 40% (Fig 2A). The results suggested that it was the combined effect of the detergent and hypotonicity that was responsible for the marked decrease in Ty2 survival. In support of this idea, Triton/water-induced killing of Ty2 was inhibited in a dose-dependent manner by the presence of NaCl at 50 and 100 mM concentrations (Fig 2B). Interestingly, when we used bacteria prepared from well-aerated overnight shaking cultures, incubation in just distilled water, either on ice or at 37°C, resulted in about 99% killing of Ty2, with no significant effect on SL1344 (Fig 2C and 2D), similar to the effect of the combination of Triton X-100 and water on Ty2 grown in static culture. Thus, Ty2 appears to be more sensitive than SL1344 to low osmolarity even in the absence of detergent, a characteristic that is particularly prominent in bacteria grown under well-aerated conditions.

To further evaluate the effects of hypotonic conditions on Ty2, we assessed growth of the bacteria at 37°C in standard LB (Miller formulation, 10 g/l NaCl, 400 mosmol/l) and in hypotonic LB (no NaCl, 100 mosml/l) under well-aerated shaking conditions [45, 46]. As shown in Fig 3A, growth of Ty2 was clearly inhibited under the hypotonic conditions during the initial 6 hours, although this effect was not seen after overnight incubation. In contrast, the growth of SL1344 was not affected by the hypotonic conditions at any stage of the analysis (Fig 3A). These results were substantiated by enumerating viable bacteria following 6 hours of growth in standard or hypotonic LB at 37°C. The experiment confirmed that there was a significant reduction in the number of viable Ty2 in the hypotonic LB, whereas there was no effect on the viability of SL1344 (Fig 3B).

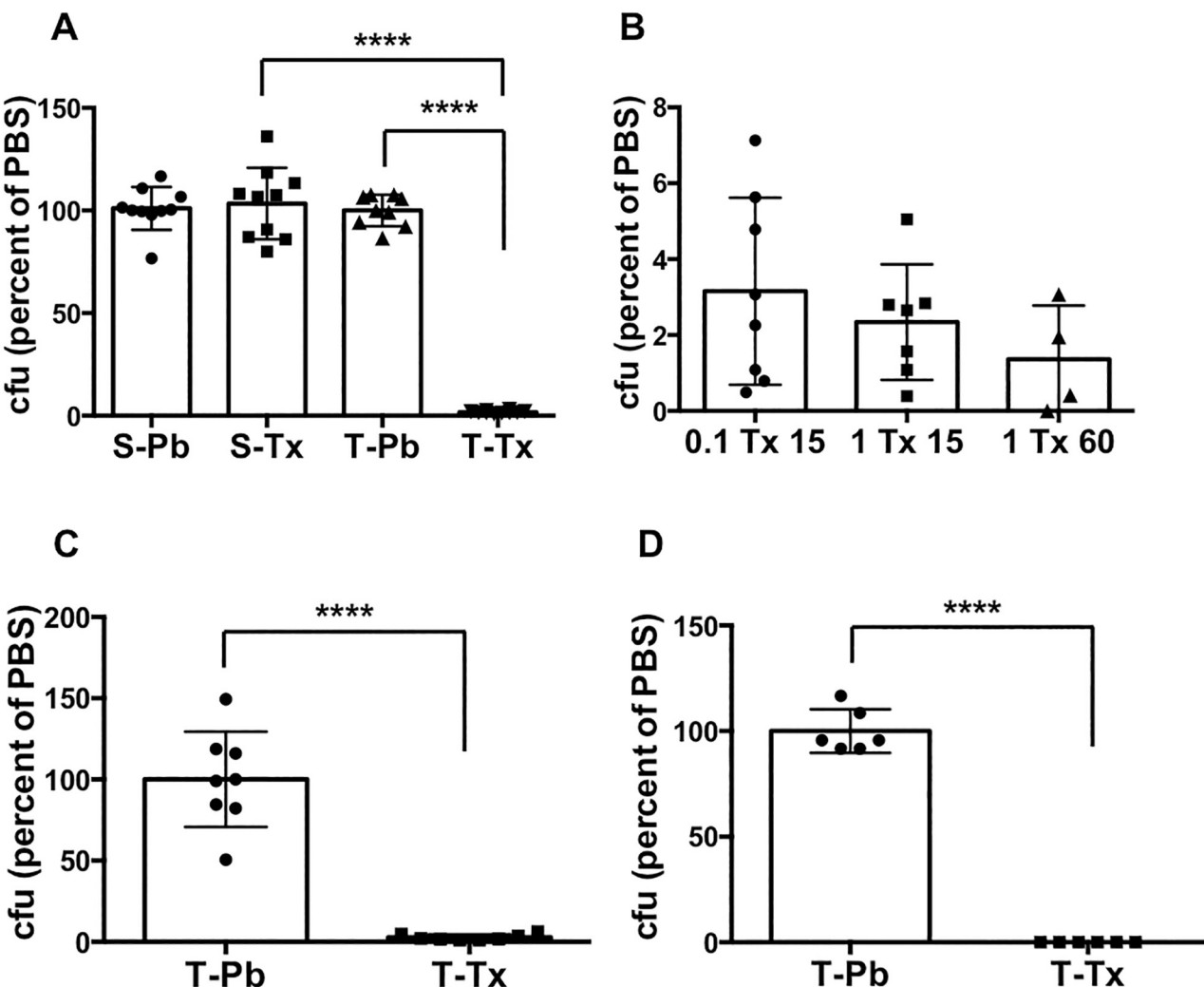

**Fig 1. Effect of Triton X-100 in water on viability of SL1344 and Ty2. A.** Equivalent numbers of SL1344 (S) or Ty2 (T) bacteria, grown overnight in static cultures, were incubated in PBS (Pb) or 1% Triton X-100 in water (Tx) for 1 hour on ice. The number of bacteria surviving in Triton was determined and expressed as a percentage of the number in PBS for each strain. ****p < 0.0001, n = 9 or 10 per group. **B.** Equivalent numbers of Ty2 bacteria, grown overnight in static cultures, were incubated in PBS or in 0.1% or 1% Triton X-100 in water (Tx) for 15 or 60 minutes on ice as indicated. The number of bacteria surviving in Triton was determined and expressed as a percentage of the number in PBS (the values for PBS are not shown for ease of visualization). n = 4–8 per group. Note the different Y axis scales in **A** and **B**. **C.** Equivalent numbers of Ty2 (T), grown overnight in shaking culture, were incubated in PBS (Pb) or in 1% Triton X-100 in water (Tx) for 60 minutes on ice. The number of bacteria surviving in Triton was determined and expressed as a percentage of the number in PBS. ****p < 0.0001, n = 8 per group. **D.** Equivalent numbers of Ty2 (T), grown overnight in shaking culture, were incubated in PBS (Pb) or in 1% Triton X-100 in water (Tx) for 60 minutes at 37˚C. The number of bacteria surviving in Triton was determined and expressed as a percentage of the number in PBS. ****p < 0.0001, n = 6 per group.

## The heightened susceptibility of *S.* Typhi Ty2 to hypotonic conditions is the result of a mutated *rpoS* gene

The Ty2 strain of *S.* Typhi is known to have a frameshift mutation in the *rpoS* gene [41, 43]. Similar inactivating mutations of *rpoS* have been documented in about a third of clinical isolates of *S.* Typhi, whereas most *S.* Typhimurium strains, including SL1344, carry the wild-type gene [49]. To determine whether the mutant *rpoS* gene might contribute to the heightened susceptibility of Ty2 to low osmolarity, we first analyzed expression of two transcriptional targets of *rpoS*: *katE*, encoding a major catalase involved in resistance to hydrogen peroxide, and *otsA*,

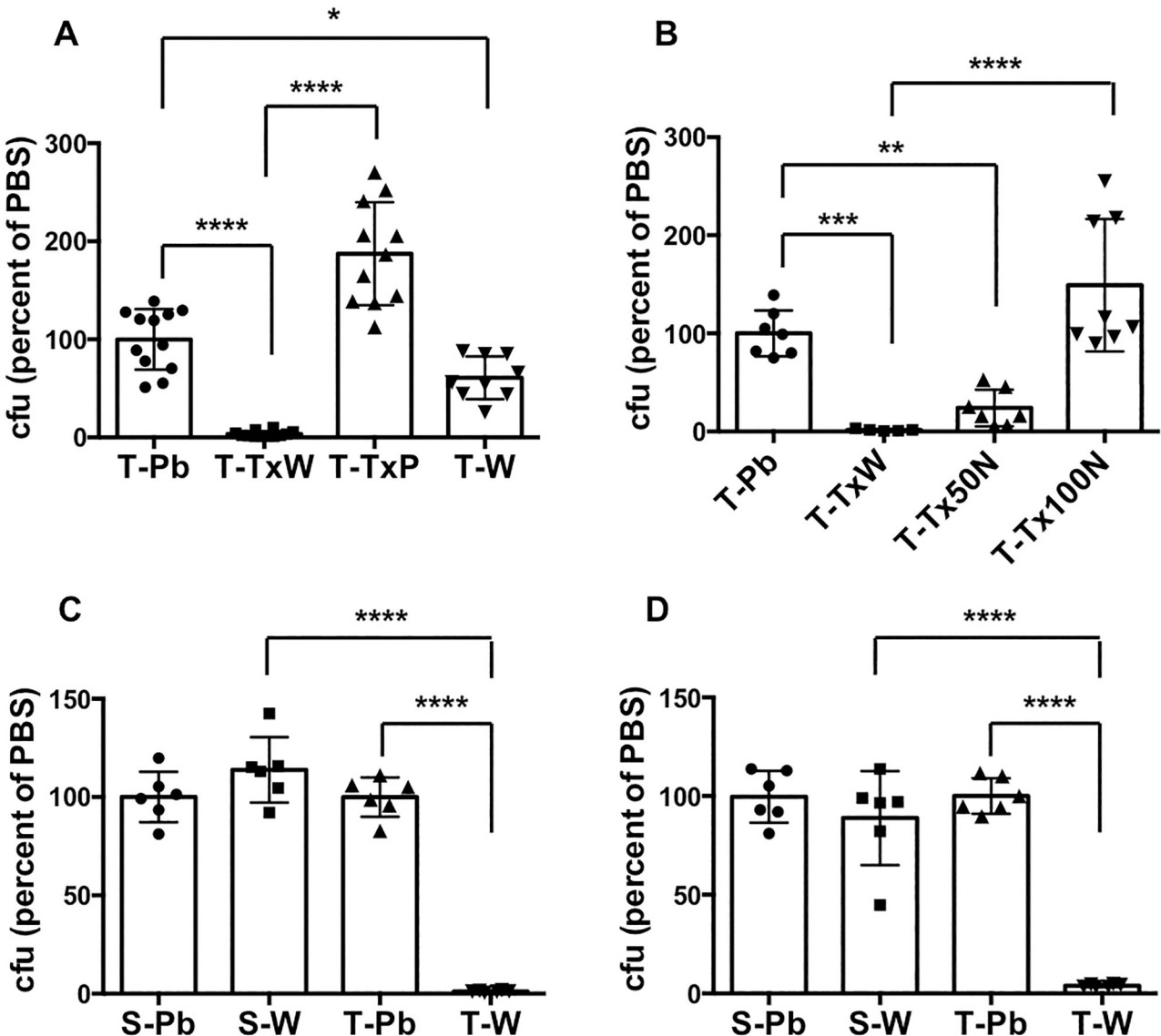

**Fig 2. Hypotonic conditions are required for the effect of Triton X-100 on Ty2. A.** Equivalent numbers of Ty2 (T), grown overnight in static culture, were incubated in PBS (Pb), in 1% Triton X-100 dissolved in either water (TxW) or PBS (TxP), or in sterile distilled water (W) for 1 hour on ice. The number of bacteria surviving in each condition was determined and expressed as a percentage of the number in PBS. *p = 0.0274, ****p < 0.0001, n = 9–12 per group. **B.** Equivalent numbers of Ty2 (T) bacteria, grown overnight in static culture, were incubated in PBS (Pb), in 1% Triton X-100 in water (TxW), or in 1% Triton X-100 (Tx) in 50 or 100 mM NaCl (N) for 1 hour on ice. The number of bacteria surviving in each condition was determined and expressed as a percentage of the number in PBS. **p = 0.0089, ***p = 0.0018, ****p < 0.0001, n = 5–8 per group. **C.** Equivalent numbers of SL1344 (S) or Ty2 (T) bacteria, grown overnight in shaking cultures, were incubated in PBS (Pb) or sterile distilled water (W) for 1 hour on ice. The number of bacteria surviving in each condition was determined and expressed as a percentage of the number in PBS for each strain. ****p < 0.0001, n = 6 per group. **D.** Equivalent numbers of SL1344 (S) or Ty2 (T) bacteria, grown overnight in shaking cultures, were incubated in PBS (Pb) or sterile distilled water (W) for 1 hour at 37˚C. The number of bacteria surviving in each condition was determined and expressed as a percentage of the number in PBS for each strain. ****p < 0.0001, n = 6 per group.

which encodes a trehalose synthase required to survive hyperosmotic stress [31, 33]. We prepared RNA from SL1344 and Ty2 isolated from overnight shaking cultures grown to stationary phase, when *rpoS* is maximally expressed, and carried out quantitative reverse transcription-PCR analysis of *katE* and *otsA* expression. As shown in Fig 4, Ty2 expressed barely detectable

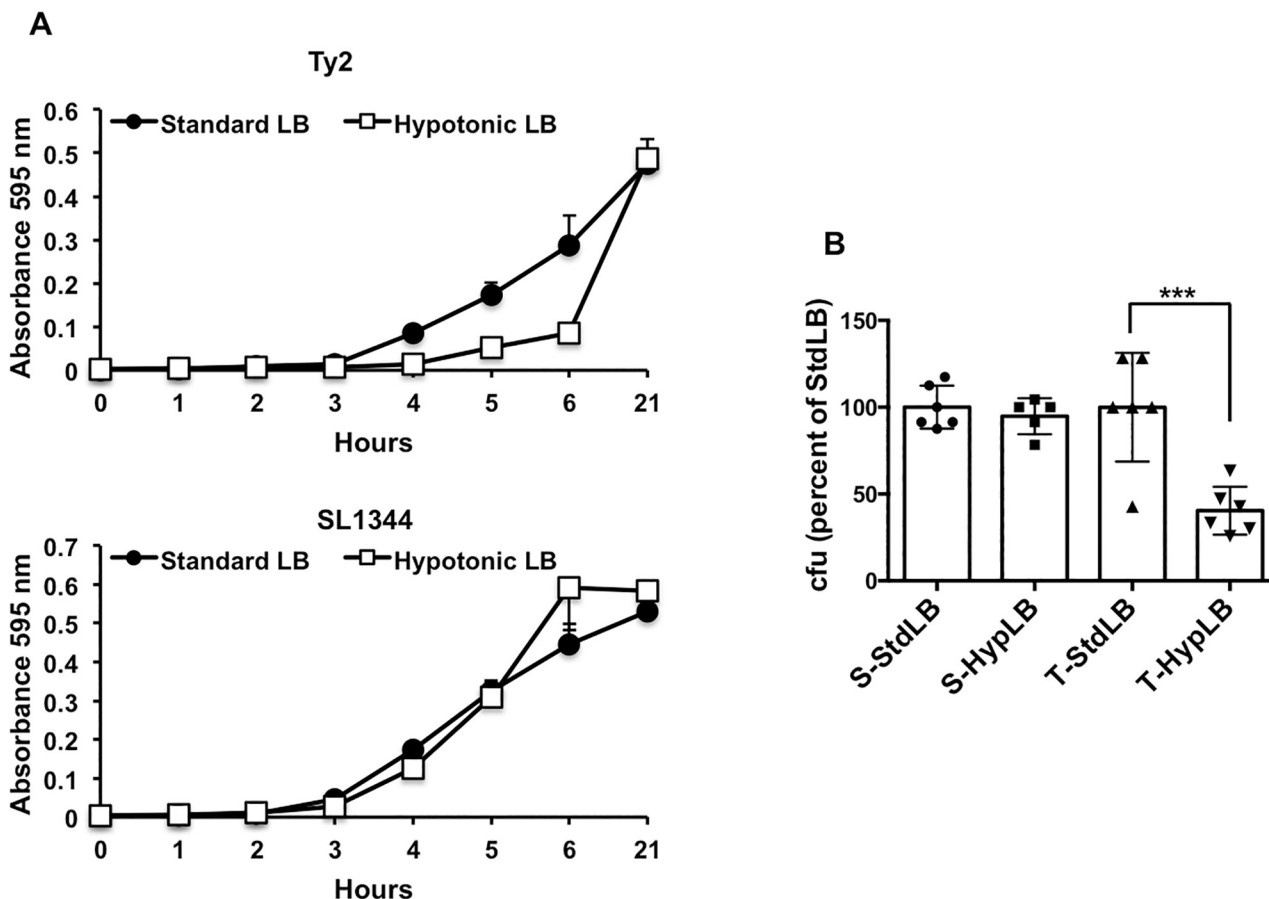

**Fig 3. Growth of Ty2, but not SL1344, is impaired by hypotonic conditions. A.** Overnight shaking cultures of Ty2 and SL1344 were diluted 1000-fold into fresh medium, either standard LB or hypotonic LB, and the bacteria were grown at 37˚C with shaking at 220 rpm. Growth curves were generated as described in the methods section. **B.** Overnight shaking cultures of Ty2 (T) and SL1344 (S) were diluted 1000-fold into fresh medium, either standard LB (StdLB) or hypotonic LB (HypLB), and the bacteria were grown at 37˚C with shaking at 220 rpm for 6 hours. The number of viable bacteria in each condition at that time point was determined and expressed as a percentage of the number in standard LB. ***p = 0.0001, n = 5 or 6 per group.

levels of both genes compared to SL1344, consistent with the absence of a functional *rpoS* in the former strain.

We then cloned the wild-type *rpoS* from SL1344 into the expression vector pBH and transformed the contruct into Ty2 to generate the strain Ty2/pBHSTm*rpoS*. As a control, we also transformed Ty2 with the empty vector to generate the strain Ty2/pBH. The growth curves of Ty2/pBHSTm*rpoS* and Ty2/pBH were very similar, although both had a longer lag phase than the parental Ty2 (Fig 5A). To confirm functional expression of *rpoS* in the transformant, we first analyzed the expression of its downstream target *katE* by quantitative reverse transcription-PCR. As shown in Fig 5B, *katE* levels in Ty2/pBHSTm*rpoS* were significantly higher (about 45-fold) than in Ty2, an effect that was not seen in Ty2/pBH. Although a 45-fold increase represents a robust over-expression of *katE* relative to Ty2, it is lower than the level seen in χ3744, an *S.* Typhi strain with a wild-type *rpoS* gene [36], which expresses about a 600-fold higher amount of *katE* than Ty2 (Fig 5B). Thus, the effect of plasmid-mediated *rpoS* expression on *katE* levels in Ty2 can be considered to be within the wild-type range. To further substantiate the functionality of *rpoS* expression and the consequent increase in *katE* levels, we assessed the ability of the transformant Ty2/pBHSTm*rpoS*

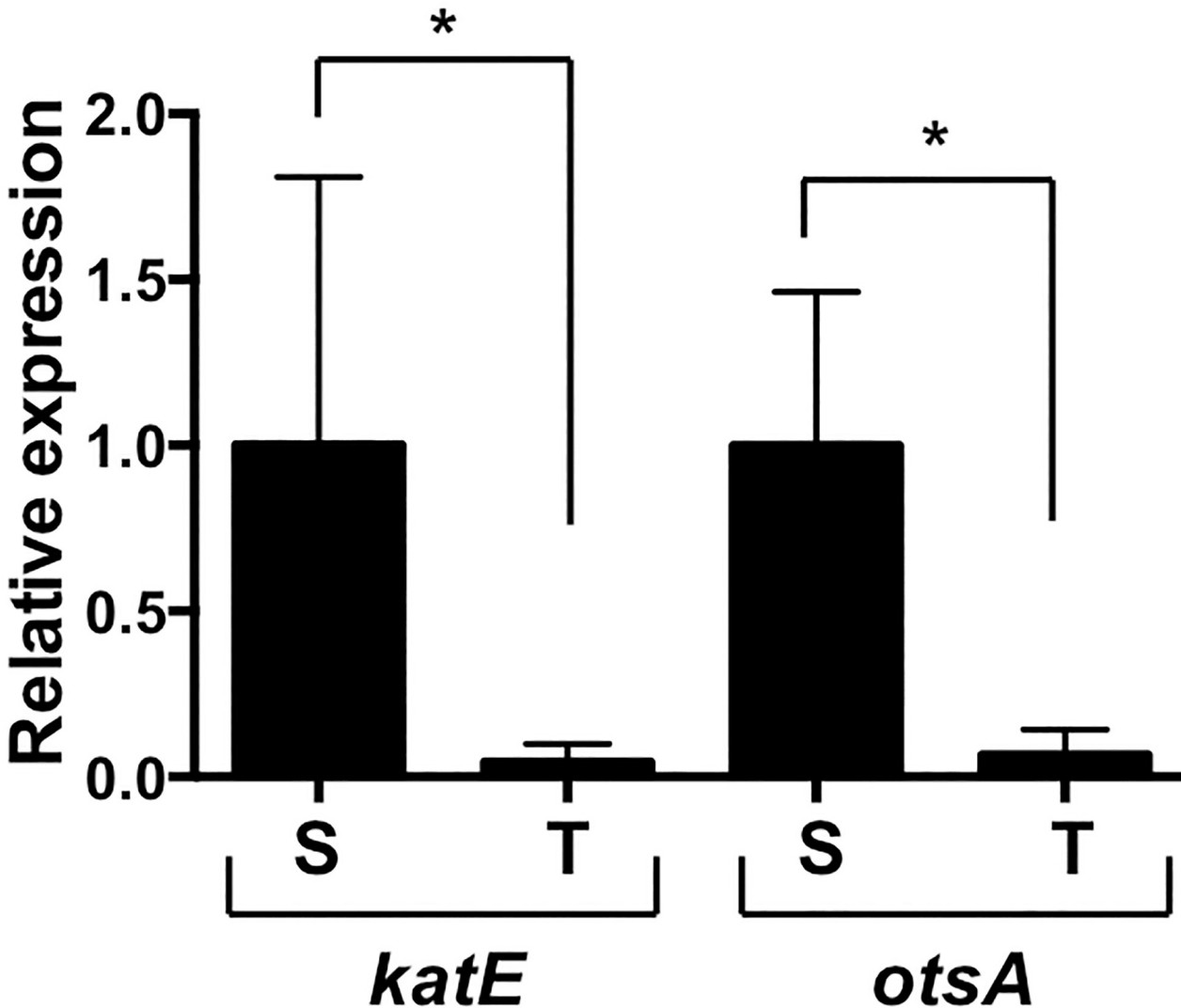

**Fig 4. Expression of *rpoS* target genes in SL1344 and Ty2.** Expression of the *katE* and *otsA* genes in overnight shaking cultures of SL1344 (S) and Ty2 (T) was determined by quantitative reverse transcription-PCR. Levels of each transcript were calculated relative to the levels in SL1344. *p = 0.02, n = 4 or 5 per group.

to survive 1 hour incubation in 0.1 mM hydrogen peroxide in PBS at 37˚C. Other investigators have shown that strains of *S*. Typhi with mutant *rpoS* are rapidly killed by hydrogen peroxide and that expression of the wild-type gene in such strains confers resistance to the oxidizing agent [43]. Consistent with these published observations and with the *rpoS*-induced up-regulation of *katE* that we have documented (Fig 5B), we found that Ty2 transformed with the *rpoS* expression construct had markedly enhanced resistance to hydrogen peroxide relative to the parental strain, whereas the empty vector had little or no effect (Fig 5C).

Next, we determined the effect of expressing wild-type *rpoS* on the viability of Ty2 in distilled water. Transformation with the wild-type *rpoS* expression construct, but not the empty vector, significantly enhanced the ability of Ty2 to survive incubation in water for 1 hour, both on ice and at 37˚C (Fig 6A and 6B). The bacteria were grown overnight in well-aerated shaking

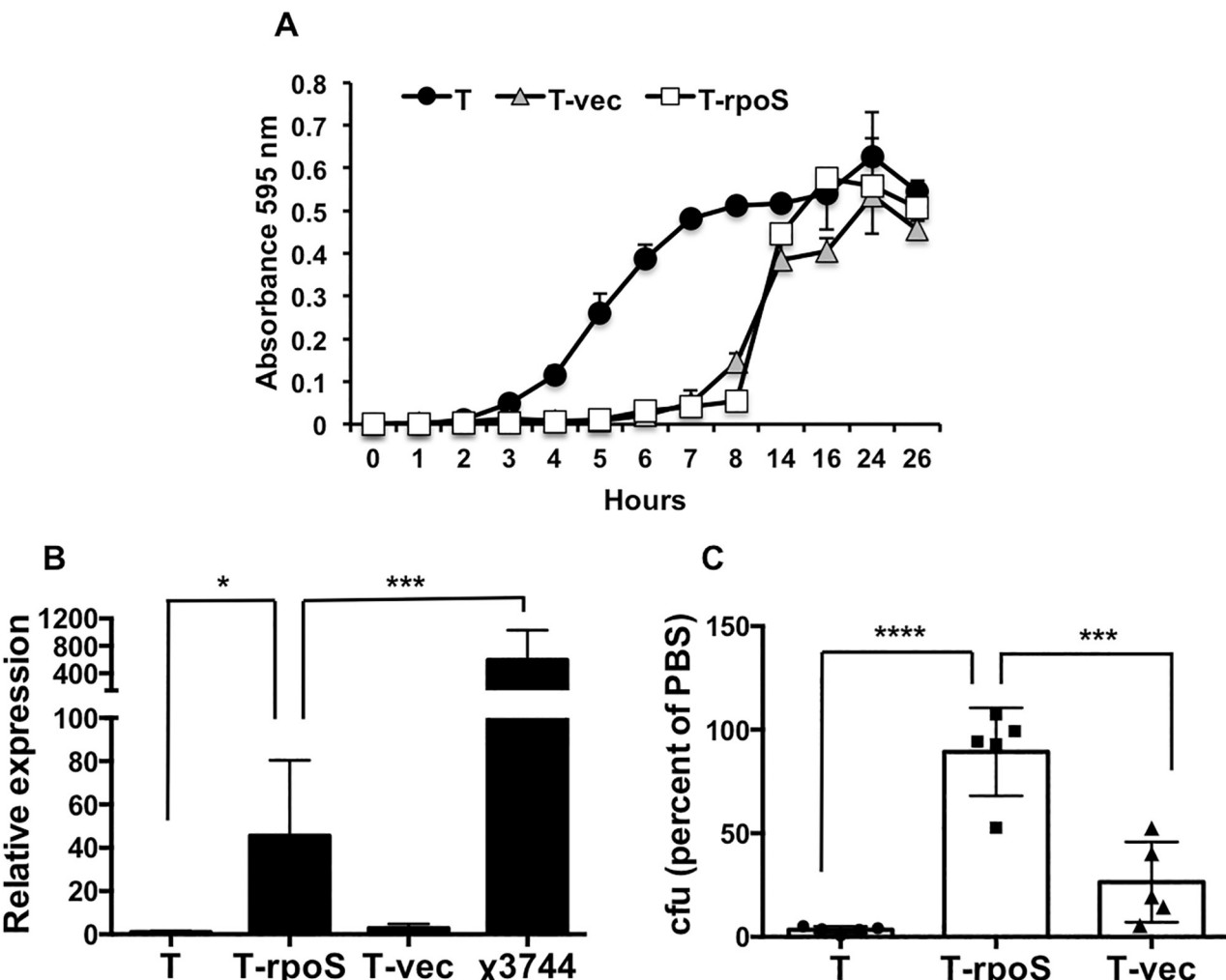

**Fig 5. Expression of wild-type *rpoS* in in Ty2. A.** Overnight shaking cultures of Ty2 (T), Ty2 transformed with wild-type *rpoS* (T-rpoS) and Ty2 transformed with empty vector (T-vec) were diluted 1000-fold into fresh LB and the bacteria were grown at 37˚C with shaking at 220 rpm. Growth curves were generated as described in the methods section. **B.** Expression of the *katE* gene in overnight shaking cultures of Ty2 (T), Ty2 transformed with wild-type *rpoS* (T-rpoS), Ty2 transformed with empty vector (T-vec) and χ3744 (wild-type *rpoS*) was determined by quantitative reverse transcription-PCR. Levels of the transcript in each strain were calculated relative to the level in Ty2. *p = 0.034, ***p = 0.0002, n = 6 per group. **C.** Equivalent numbers of Ty2 (T), Ty2 transformed with wild-type *rpoS* (T-rpoS) and Ty2 transformed with empty vector (T-vec), grown overnight in shaking cultures, were incubated in PBS or PBS containing 0.1 mM hydrogen peroxide for 1 hour at 37˚C. The number of bacteria surviving in hydrogen peroxide was determined and expressed as a percentage of the number in PBS for each strain (the values for PBS are not shown for ease of visualization). ***p = 0.0001, ****p < 0.0001, n = 5 per group. The difference between T and T-vec was not statistically significant.

cultures for these experiments. The results suggest that the marked sensitivity of Ty2 to hypotonic conditions (as represented by water) may be attributable to the mutation in the *rpoS* gene.

Although these findings implicate *rpoS* in resistance to low osmolarity conditions, they leave open the formal possibility that the plasmid-directed over-expression of *rpoS* in Ty2 could compensate for some other factor that makes the strain sensitive to a hypotonic environment. Therefore, to demonstrate the role of *rpoS* more conclusively, we compared the ability of the *S.* Typhi ISP1820 strains χ3744 (wild-type *rpoS*), χ9061 (inactive, mutated *rpoS*), χ9066 (wild-type *rpoS* allele replaced with arabinose-inducible wild-type *rpoS*) to survive incubation in distilled water for 1 hour at 37˚C. The bacteria were prepared from overnight shaking cultures, with χ9066 grown in the absence or presence of 0.2% arabinose. As shown in Fig 7, the

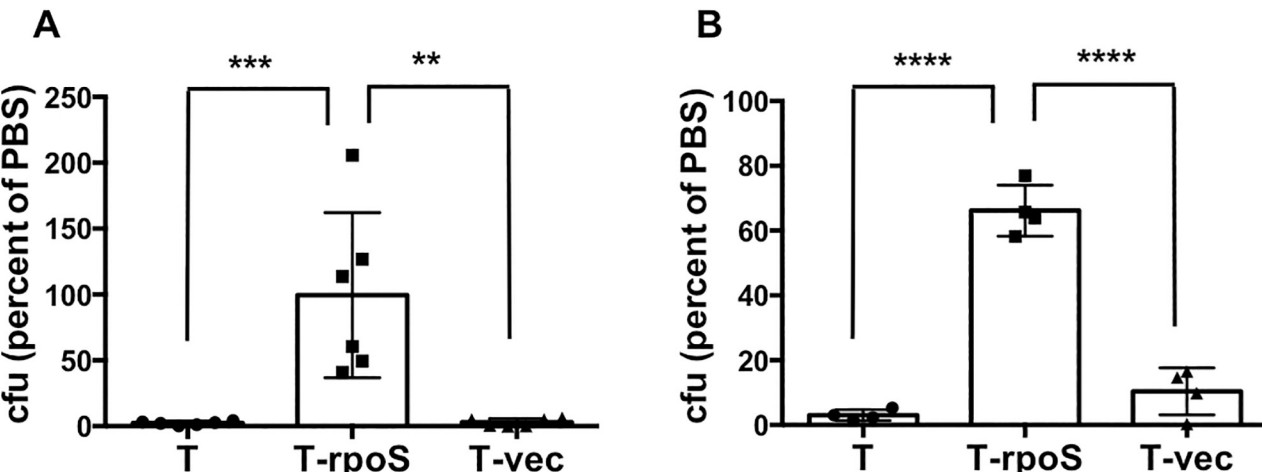

**Fig 6. Wild-type *rpoS* confers resistance to water in Ty2.** Equivalent numbers of the parental Ty2 (T), Ty2 transformed with wild-type *rpoS* (T-rpoS) and Ty2 transformed with empty vector (T-vec), grown overnight in shaking cultures, were incubated in PBS or in sterile distilled water for 1 hour on ice (**A**) or at 37°C (**B**). The number of bacteria surviving in water was expressed as a percentage of the number in PBS for each strain (the values for PBS are not shown for ease of visualization). **A**, **p = 0.0007, ***p = 0.0006, n = 6 per group. **B**, ****p < 0.0001, n = 4 per group.

survival of χ3744 was not significantly affected by incubation in water (unlike our results with Ty2), while the viability of χ9061 under these conditions was significantly reduced and that of χ9066 was restored to wild-type levels only if *rpoS* expression was induced by culture in arabinose. These results confirm that *rpoS* is required for *S.* Typhi to survive in the low osmolarity environment of distilled water. Interestingly, the effect of the *rpoS* mutation on survival in water was less marked in χ9061 than in Ty2 –viability of χ9061 after 1 hour in water at 37°C was about 30% of that in PBS, whereas the viability of Ty2 under the same conditions was about 4% (compare the results in Figs 2D and 7, p < 0.0001). Presumably, this difference reflects variations in the genetic backgrounds of the 2 strains.

## Discussion

The results of our experiments indicate that the viability of the widely used *S.* Typhi strain Ty2 is markedly reduced by exposure to hypotonic conditions, in contrast to the *S.* Typhimurium strain SL1344, which survives as well in distilled water as in PBS. This difference between the strains correlates with the presence of a mutated, non-functional *rpoS* gene in Ty2 and a wild-type version of the gene in SL1344. We demonstrated the functional relevance of this correlation by showing that expression of the wild-type *rpoS* in Ty2 increased survival in water to the level of SL1344. Furthermore, we confirmed the role of *rpoS* by showing that an *S.* Typhi strain with wild-type *rpoS* retained normal viability in water, that inactivation of the gene in this strain significantly reduced survival in water and that the latter abnormality could be corrected by inducing expression of wild-type *rpoS*. Taken together, our results indicate that *rpoS* plays an important role in maintaining the viability of *S.* Typhi (and presumably of *S.* Typhimurium also) in a low osmolarity environment. This function adds to the previously documented involvement of *rpoS* in the ability of *Salmonella* to adapt to other stresses such as hyperosmolarity, acidity, nutrient deprivation and low temperature, and to cause lethal infection in mice [26–29, 34, 35, 43, 50, 51].

The details of how exactly *rpoS* protects against low osmolarity in *S.* Typhi remain to be elucidated. However, studies carried out in *E. coli* provide clues about the ways in which bacteria

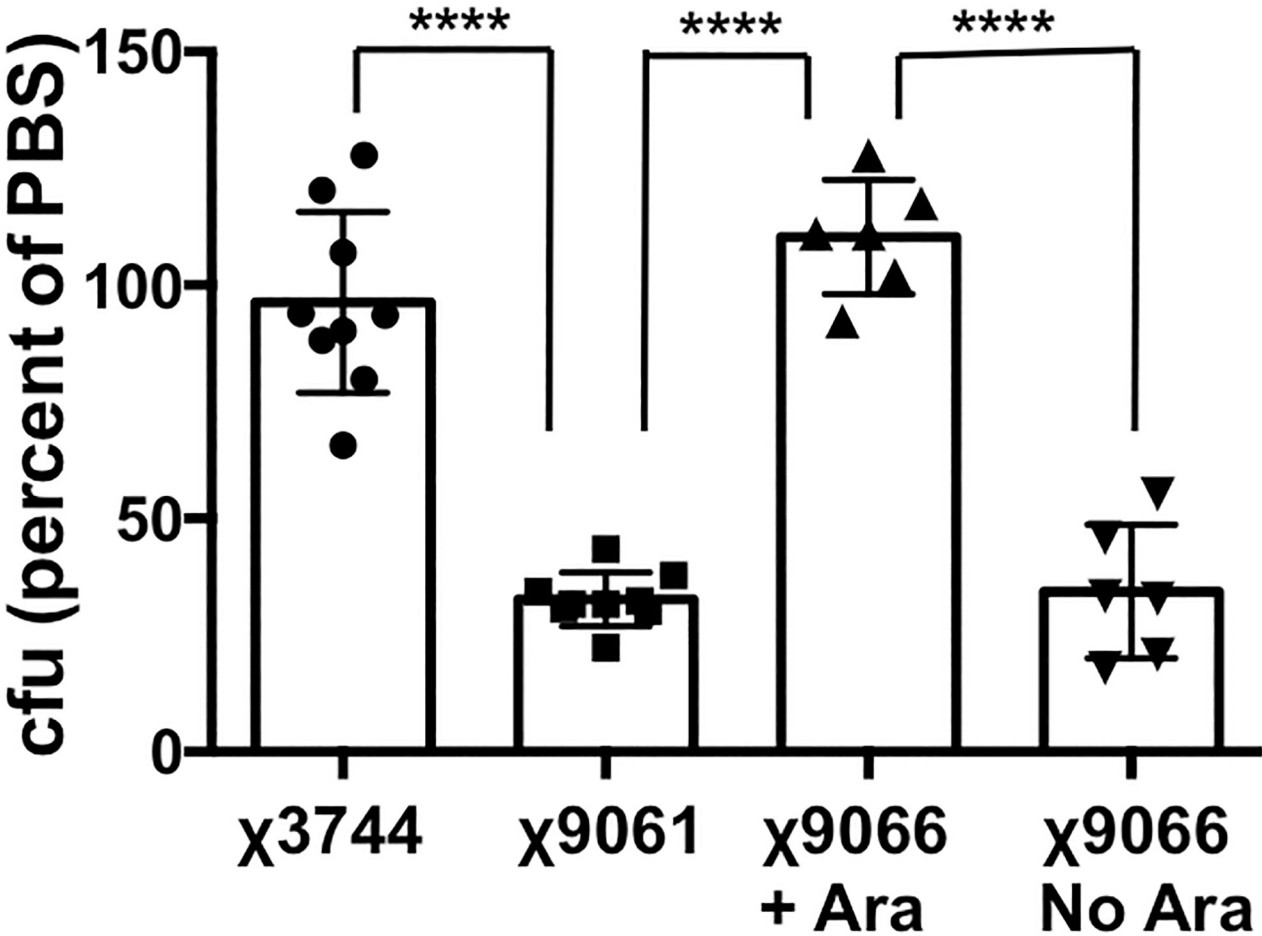

**Fig 7. *rpoS* is required for survival of *S.* Typhi in water.** Equivalent numbers of the *S.* Typhi strains χ3744 (wild-type *rpoS*), χ9061 (*rpoS* inactivated), χ9066 (arabinose-inducible *rpoS*), grown overnight in shaking cultures (with our without added arabinose–Ara–in the case of χ9066, as indicated), were incubated in PBS or in sterile distilled water for 1 hour at 37˚C. The number of bacteria surviving in water was expressed as a percentage of the number in PBS for each strain (the values for PBS are not shown for ease of visualization). ****$p < 0.0001$, n = 6–9 per group.

adapt to an osmotic downshift [52]. A decrease in the osmolarity of the environment causes a rapid influx of water into the bacterial cell, leading to an increase in outward turgor pressure that has the potential to disrupt the cell envelope and cause death. Bacteria are able to resist or counteract these events by alterations in cytoplasmic solute content and changes to the composition and mechanical properties of cellular membranes and the peptidoglycan layer. An important aspect of this adaptive response involves mechanosensitive channels in the cytoplasmic membrane, exemplified by MscS and MscL, which become activated as a result of an increase in membrane tension, allowing the rapid release of cytoplasmic solutes and the consequent relief of turgor pressure [53–55]. Adequate levels of these channel proteins are required for bacteria to survive a significant decrease in external osmolarity [56]. Interestingly, experiments in *E. coli* have shown that the expression of both MscS and MscL increases in stationary phase in an *rpoS*-dependent manner [54]. Correspondingly, and in keeping with the findings reported here, an *rpoS* null mutant of *E. coli* undergoes rapid and dramatic lytic death when shifted from stationary phase culture to sterile distilled water [54]. *rpoS* has also been implicated in remodeling of the cell wall in stationary phase through its control of genes involved in

peptidoglycan synthesis and recycling [57, 58]. Thus, effects on the levels of mechanosensitive channels and on peptidoglycan biosynthesis offer potential mechanisms by which *rpoS* could affect susceptibility to low osmolarity conditions in *S*. Typhi. Whether such mechanisms actually operate in *Salmonella* during the transition to water requires further investigation.

Our experiments indicate that the loss of *S*. Typhi Ty2 viability on incubation in water is significantly less when the bacteria are cultured under static conditions than when they are grown in well-aerated shaking cultures (about 40% killing–Fig 2A–versus about 99% killing–Fig 2C–respectively, p < 0.0001). A potential explanation for this difference is that the static culture had not reached full saturation at the time the bacteria were collected for analysis–it has been shown previously that the lytic effects of an osmotic downshift on an *rpoS* mutant of *E. coli* occur only at saturation [54]. An alternative possibility is that growth under poorly aerated conditions induces or facilitates the operation of *rpoS*-independent mechanisms that confer some degree of protection against lysis in water. Such mechanisms may involve genes that have been implicated in the survival of *S*. Typhi Ty2 when the bacteria are transferred from LB broth to distilled water [24]. These genes play roles in glycogen biosynthesis and degradation, production of lipopolysaccharide, and, perhaps most pertinently, in peptidoglycan biosynthesis, but how exactly they contribute to resistance against hypoosmotic stress is not yet clear.

Even though Ty2's susceptibility to lysis in water is reduced by growth in static culture, the bacteria remain exquisitely sensitive to killing by the combined effects of hypotonicity and Triton X-100 even under these conditions. Presumably, the hypotonic environment and the absence of a functional *rpoS* result in excessive turgor pressure, which may not be lethal on its own (because of the effects of the static culture) but may make the bacteria particularly vulnerable to membrane disrupting agents like Triton. It is important to emphasize that our studies demonstrate that Triton does not affect viability of Ty2 under isotonic conditions, explaining why both *S*. Typhi and *S*. Typhimurium can grow in the presence of bile salts, which are detergent-like molecules, at concentrations of up to 3% in LB, and why *S*. Typhi can colonize the gallbladder in some patients with typhoid [59, 60].

*S*. Typhi and *S*. Typhimurium are enteropathogens that are transmitted via contaminated food and water. As such, they are exposed to environments, both within and outside their hosts, that present potentially stressful challenges, including variations in osmolarity. The transition from feces to water, a key step in the infectious cycle, is particularly pertinent to the present discussion since it represents an abrupt osmotic downshift. Our results indicate that *rpoS* is likely to play an important role in surviving this transition. In addition, since *S*. Typhi does not productively infect animals other than humans, it may have to exist for extended periods in wells, ponds, rivers and other freshwater collections that represent sources of infection in parts of the world where typhoid is endemic [20–23]. Thus, *S*. Typhi may be particularly dependent on *rpoS* for survival in such environments, raising the possibility that targeting this gene or its downstream functions could represent a strategy for interfering with transmission of the pathogen.

Given our findings, it is surprising that about one-third of *S*. Typhi clinical isolates have a mutated, functionally inactive *rpoS*, whereas most *S*. Typhimurium clinical isolates have a wild-type *rpoS* [49]. This variation in the occurrence of *rpoS* mutations between the serovars suggests that *S*. Typhi may be less dependent on *rpoS* during infection of humans than *S*. Typhimurium, and so may be able to tolerate loss of function of this gene in the host environment. It will be interesting in this context to determine the frequency of *rpoS* mutations in *S*. Typhi isolates obtained from environmental water sources. Our results predict that such mutations would be seen infrequently, if at all, in environmental freshwater isolates.

Finally, it is pertinent to mention that the *S*. Typhi Ty2 strain is widely used in laboratory studies of host-pathogen interactions, including in gentamicin protection assays that require

the use of Triton or similar detergent to lyse mammalian cells infected with the bacteria [48]. Our studies indicate that if the detergent is made in a hypotonic buffer, the number of recovered Ty2 may be falsely low.

## Supporting information

**S1 Data. Viability assay data for Fig 1A.**
(XLSX)

**S2 Data. Viability assay data for Fig 1B.**
(XLSX)

**S3 Data. Viability assay data for Fig 1C.**
(XLSX)

**S4 Data. Viability assay data for Fig 1D.**
(XLSX)

**S5 Data. Viability assay data for Fig 2A.**
(XLSX)

**S6 Data. Viability assay data for Fig 2B.**
(XLSX)

**S7 Data. Viability assay data for Fig 2C.**
(XLSX)

**S8 Data. Viability assay data for Fig 2D.**
(XLSX)

**S9 Data. Growth curve data for Fig 3A.**
(XLSX)

**S10 Data. Viability assay data for Fig 3B.**
(XLSX)

**S11 Data. Quantitative reverse transcription PCR data for Fig 4.**
(XLSX)

**S12 Data. Growth curve data for Fig 5A.**
(XLSX)

**S13 Data. Quantitative reverse transcription PCR data for Fig 5B.**
(XLSX)

**S14 Data. Viability assay data for Fig 5C.**
(XLSX)

**S15 Data. Viability assay data for Fig 6A.**
(XLSX)

**S16 Data. Viability assay data for Fig 6B.**
(XLSX)

**S17 Data. Viability assay data for Fig 7.**
(XLSX)

## Acknowledgments

We would like to thank Dr. Roy Curtiss, III and Dr. Soo-Young Wanda of the University of Florida, Gainesville for graciously sharing their *S.* Typhi strains.

## Author Contributions

**Conceptualization:** Bobby J. Cherayil.

**Data curation:** Bobby J. Cherayil.

**Formal analysis:** Bobby J. Cherayil.

**Funding acquisition:** Bobby J. Cherayil.

**Investigation:** Eamon Gibbons, Mehbooba Tamanna.

**Methodology:** Eamon Gibbons, Mehbooba Tamanna.

**Project administration:** Bobby J. Cherayil.

**Supervision:** Bobby J. Cherayil.

**Writing – original draft:** Bobby J. Cherayil.

**Writing – review & editing:** Eamon Gibbons, Mehbooba Tamanna, Bobby J. Cherayil.

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
