## [Decision Letter · Decision Letter 0]

10 Oct 2022

PONE-D-22-26135The rpoS gene confers resistance to low osmolarity conditions in Salmonella enterica serovar TyphiPLOS ONE

Dear Dr. Cherayil,

Thank you for submitting your manuscript to PLOS ONE. After careful consideration, we feel that it has merit but does not fully meet PLOS ONE’s publication criteria as it currently stands. Therefore, we invite you to submit a revised version of the manuscript that addresses the points raised during the review process.

Both reviewers are positive about the study, however, they also have some valid suggestions. In particular, both felt the introduction lacked crucial background information and that additional experiments are required to confirm the conclusions. I agree that this would significantly improve the paper. If you do resubmit, please clearly justify the omission of any of the suggested experiments in your response to the reviewers.

We look forward to receiving your revised manuscript.

Kind regards,

Olivia Steele-Mortimer, Ph.D.

Academic Editor

PLOS ONE

Journal Requirements:

Reviewers' comments:

Reviewer's Responses to Questions

**Comments to the Author**

1. Is the manuscript technically sound, and do the data support the conclusions?

Reviewer #1: Partly

Reviewer #2: Partly

2. Has the statistical analysis been performed appropriately and rigorously? 

Reviewer #1: Yes

Reviewer #2: Yes

3. Have the authors made all data underlying the findings in their manuscript fully available?

Reviewer #1: Yes

Reviewer #2: Yes

4. Is the manuscript presented in an intelligible fashion and written in standard English?

Reviewer #1: Yes

Reviewer #2: Yes

5. Review Comments to the Author

Reviewer #1: This research describes the observation that a commonly used laboratory strain of Salmonella enterica serovar Typhi, Ty2, is hypersensitive to hypotonic conditions. The authors show that unlike the Salmonella enterica serovar Tyhphimurium strain SL1344, hypotonic conditions reduce viability of Ty2 and increase sensitivity to the detergent Triton X-100. They hypothesize that the increased sensitivity is due to the known mutation in rpoS in Ty2. They transfer the wildtype rpoS gene from SL1344 to Ty2 and observe increased resistance to hypotonic conditions and conclude that rpoS confers resistance to low osmolarity conditions in S. Typhi.

The manuscript is well-written and is a good start to an interesting study of the role of RpoS in resisting hypotonic conditions. However, some additional information and experiments are needed for clarification and to fully support their conclusions.

Major

1. Please provide more information in the manuscript about plasmid pBH in addition to the reference provided, such as the promoter and copy number. Is the plasmid borne rpoS gene controlled with the rpoS native promoter or is it driven by a constitutive promoter?

2. The given reference for pBH suggests that rpoS expression in this study is driven by the trp-lac promoter. Why did the authors choose to use constitutive expression for this study? The authors need to careful about the conclusions and the title because this study is solely testing whether constitutive expression of rpoS can confer resistance to low osmolarity.

3. Please provide growth curves for strains containing empty plasmid and the rpoS expression plasmid. Constitutive expression, in general, often results in overexpression and/or expression at inappropriate times during the growth phase compared to wildtype. If there is a detrimental effect on growth, then overnight growth of the rpoS expressing strain under shaking conditions may have not reached saturation (which the authors state could affect susceptibility or resistance to hypotonic conditions). Alternatively, aberrant expression of large amounts of a protein, especially regulatory, could lead to activation of other stress pathways that could result in a similar resistance phenotype. This would not be seen in the empty plasmid control and could lead to misleading conclusions.

4. Please provide evidence for expression of rpoS in the strain containing the rpoS expression plasmid. On lines 307-308, the authors claim to have shown that expression of rpoS in Ty2 increased survival in water. Currently, the only evidence that the gene is expressed is a phenotypic readout of reduced sensitivity to hydrogen peroxide. Evidence of rpoS gene expression either by qRT-PCR (of rpoS itself and/or selected rpoS target genes) or with Western blotting is needed to support these claims. Is the expression level comparable to native expression under the experimental conditions?

Minor

1. Please include additional information about tube size and volumes used for culture growth. For growth curves, what volume was taken out of the culture for absorbance readings?

2. Please provide more information about the qRT-PCR analysis. Were the qRT-PCR primer efficiencies calculated and equivalent?

3. In Figure 5, why does the presence of the empty vector confer some resistance to hydrogen peroxide? Is this statistically significant?

Reviewer #2: The authors present a study in which the Salmonella Typhi strain Ty2 is shown to be sensitive to low osmolarity and that this can be reversed by introducing a cloned WT rpoS gene into the strain. The study is well-presented with clear logic flow and associated figures. However, I do have specific comments below that should be addressed.

1. The introduction needs more discussion of both the rpoS gene itself in relation to environmental stress resistance and the state of past rpoS research in Salmonella Typhi. I realize that a full discussion of rpoS would be large (potentially beyond scope of this introduction), but the reader should have some more information about this gene than what is currently present. For the S. Typhi rpoS literature, the labs of Norel and Curtiss have significant contributions to this area and should be discussed (both in introduction and discussion).

2. Line 56-58. This sentence should be cited with references – there is a large body of work that established this.

3. Sentence ending in line 61 should also have more references beyond just a single review since many studies have worked in this area.

4. The “nail in the coffin” for this study would have been to take a WT, low-osmolarity resistant S. Typhi strain and mutate that rpoS gene and then show similar loss in resistance as seen in Ty2. And then, complement this mutant with the cloned WT rpoS. This experiment would significantly add to this study.

5. The authors should more fully discuss the possibility that the low osmolarity sensitivity in Ty2 could be due to some other factor than rpoS, and that the cloned/overexpressed rpoS is just compensating for that other factor. It would also be helpful to introduce the cloned rpoS gene into other S. Typhi rpoS mutant, low osmolarity sensitive strains and confirm that this reversed the sensitivity in the same manner (or not). In addition, it would be informative to test the S. Typhi strains/isolates with WT rpoS genes for low osmolarity resistance – do they all display this, as predicted? If not, then maybe there is some other rpoS-independent factor at play. But if so, then that really bolsters the rpoS model presented in this paper.

6. PLOS authors have the option to publish the peer review history of their article (what does this mean?). If published, this will include your full peer review and any attached files.

Reviewer #1: No

Reviewer #2: No

---

## [Author Response · Author response to Decision Letter 0]

15 Nov 2022

My co-authors and I would like to thank the editors and reviewers for their careful evaluation of our manuscript, and for their thoughtful comments and suggestions. We have carried out the key experiments suggested by the reviewers and have incorporated the results into the revised manuscript. The new findings have helped significantly to strengthen the overall conclusions of the study. We hope that the work will now be considered suitable for publication in PLoS One. 

 We have responded to each of the points mentioned by the reviewers as indicated below. The page and line numbers mentioned in the responses refer to the marked-up manuscript file. 

Reviewer 1

 We are gratified that the reviewer found the manuscript to be “well-written” and “a good start to an interesting study of the role of RpoS in resisting hypotonic conditions”.

Major

1. Please provide more information in the manuscript about the plasmid pBH….

 In response to the reviewer’s request, we have now provided more information about pBH in the Methods section of the revised manuscript (p. 8, lines 236-239), including promoter (trp-lac) and copy number (high copy). Since the S. Typhi Ty2 strain does not express the lac repressor, the promoter functions in a constitutive fashion.

2. Why did the authors choose to use constitutive expression for this study? The authors need to be careful about the conclusions and title because this study is solely testing whether constitutive expression of rpoS can confer resistance to low osmolarity. 

 We chose pBH-mediated constitutive expression of rpoS because our earlier work with this plasmid yielded robust and reliable expression in Salmonella (Infect. Immun. 2000; 68: 5567). This has been explained on p. 8, lines 238-239 of the revised manuscript. The reviewer raises a valid concern that the results with this system as reported in the original version of the manuscript merely show that constitutive expression of rpoS can confer resistance to hypotonic conditions. However, we have now carried out additional experiments using S. Typhi strains kindly provided by Dr. Roy Curtiss III that substantiate and extend our findings with plasmid-directed rpoS expression in Ty2. In these new experiments, we assessed the survival in distilled water of the strains χ3744 (S. Typhi ISP1820, wild-type rpoS gene), χ9061 (S. Typhi ISP1820, rpoS inactive allele expressing a truncated protein) and χ9066 (S. Typhi ISP1820, arabinose-inducible wild-type rpoS). As shown in Fig. 7 and described on p. 15-16, lines 451-472 of the revised manuscript, the viability of χ3744 was not compromised by incubation in water, in contrast to the rpoS-mutant strain Ty2. The rpoS-inactive strain χ9061 had significantly reduced viability in water, an abnormality that was corrected in χ9066 only when the bacteria were grown in the presence of arabinose. These new findings confirm the overall conclusions of the study and the appropriateness of the title. 

3. Please provide growth curves for strains containing empty plasmid and the rpoS expression plasmid…..

 To address the reviewer’s concerns, we have now provided growth curves for the parental Ty2 strain along with the strains containing empty plasmid (Ty2/pBH) and the rpoS expression plasmid (Ty2/rpoS) (Fig. 5A of the revised manuscript, described on p. 13, lines 387-388). The data indicate that there is no difference between the growth patterns of the latter 2 strains, although they both have a longer lag phase than the parental Ty2. Importantly, both of the plasmid-containing strains reach saturation at about the same time, including the time at which the cultures are harvested for analysis. Thus, it is unlikely that differences in growth phase at the time of analysis can explain the difference in susceptibility to hypotonic conditions between Ty2/pBH and Ty2/rpoS. Moreover, our new experiments with the S. Typhi strains from Dr. Curtiss clearly demonstrate that the viability of a strain with wild-type rpoS is not affected by incubation in distilled water, and that inactivation of rpoS in this strain significantly compromises viability under these conditions (Fig. 7 of the revised manuscript). These findings confirm the role of rpoS in resisting low osmolarity, and also indicate that our results with plasmid-directed expression of rpoS in Ty2 are unlikely to reflect the effects of over-expression or inappropriate expression. 

4. Please provide evidence for expression of rpoS in the strain containing the rpoS expression plasmid…..Evidence of rpoS gene expression by qRT-PCR (of rpoS itself and/or selected rpoS target genes)….is needed… 

 To address the reviewer’s concern, we have carried out qRT-PCR analysis to examine expression of the rpoS target gene katE. The results (Fig. 5B of the revised manuscript, described on p. 13-14, lines 389-402) indicate that expression of katE is about 45-fold higher in the strain containing the rpoS expression plasmid (Ty2/pBHSTmrpoS) relative to the parental Ty2 strain. As requested by the reviewer, we also examined katE expression in an S. Typhi strain that has a wild-type rpoS gene (χ3744), and found it to be about 600-fold higher than in Ty2. Thus, the effect of plasmid-mediated rpoS expression on katE levels in Ty2 can be considered to be within the wild-type range. 

Minor

1. Please include additional information about tube size and volumes used for culture growth….

 Bacterial strains were grown at 37oC in sterile 12 ml polypropylene tubes, using 2 ml of LB in loosely capped tubes or 10 ml of LB in tightly capped tubes for shaking or static cultures, respectively. For generation of growth curves, shaking cultures were carried out in triplicate in sterile 24-well tissue culture plates with 1.2 ml of LB per well. 100 µl of the culture was removed from each well at each time point to measure absorbance in a microplate reader. To ensure that the culture volume in the wells was not depleted, separate plates were used for the measurements at 0-6 hr, 7-16 hr and 24-26 hr. As requested by the reviewer, we have now provided these details in the Methods section (p. 5, lines 132-139, and p. 6, lines 159-169 of the revised manuscript).

2. Please provide more information about the qRT-PCR analysis….

 As requested by the reviewer, we have provided a more detailed description of the various steps of the qRT-PCR analysis, including temperatures and times of incubation, reaction component concentrations and amplication conditions (Methods section, p. 7, lines 190-212 of the revised manuscript). Sequences of primers were designed using Primer-BLAST and their specificity was confirmed by melting curve analysis of the amplification products. All products yielded single, sharp melting curve peaks. 

3. In Figure 5, why does the presence of the empty vector confer some resistance to hydrogen peroxide? Is this statistically significant?

 Even though the peroxide resistance of the strain with the empty vector appears to be greater than the parental Ty2 strain, the difference is not statistically significant and presumably reflects experimental variation. We have mentioned this in the revised legend to Fig. 5 (p. 15, line 433 of the revised manuscript).

Reviewer 2

 We are pleased that the reviewer found the study to be “well-presented with clear logic flow and associated figures”. The specific comments made by the reviewer have been addressed below.

1. The introduction needs more discussion of the rpoS gene….

 As requested by the reviewer, we have expanded the Introduction section of the manuscript to present more information about Salmonella rpoS and have cited the relevant papers, including key publications from the Curtiss and Norel labs, in both the Introduction and Discussion (p. 4-5, lines 94-115, and p. 17, lines 497-498 of the revised manuscript).

2. Line 56-58. This sentence should be cited with references….

 We have cited the relevant references in support of this statement (p. 3, lines 65-67 of the revised manuscript). 

3. Sentence ending in line 61 should also have more references….

 We have supported this statement with additional references beyond the review originally cited (p. 3, lines 67-70 of the revised manuscript).

4. The “nail in the coffin” for this study would have been to take a WT, low osmolarity resistant S. Typhi strain and mutate that rpoS gene…..

 Thank you for this suggestion. We have carried out the exact experiment proposed by the reviewer using S. Typhi strains kindly provided by Dr. Roy Curtiss III. In this new analysis, we assessed the survival in distilled water of the strains χ3744 (S. Typhi ISP1820, wild-type rpoS gene), χ9061 (S. Typhi ISP1820, rpoS inactive allele expressing a truncated protein) and χ9066 (S. Typhi ISP1820, arabinose-inducible wild-type rpoS). As shown in Fig. 7 and described on p. 15-16, lines 451-472 of the revised manuscript, the viability of χ3744 was not compromised by incubation in water, in contrast to the rpoS-mutant strain Ty2. The rpoS-inactive strain χ9061 had significantly reduced viability in water, an abnormality that was corrected in χ9066 only when the bacteria were grown in the presence of arabinose. These results are consistent with our findings with plasmid-directed expression of rpoS in Ty2, and confirm the idea that rpoS is required to resist low osmolarity conditions.

5. The authors should more fully discuss the possibility that the low osmolarity sensitivity in Ty2 could be due to some other factor than rpoS….. 

 Our new findings with the strains obtained from Dr. Curtiss (Fig. 7) indicate clearly that rpoS is necessary for normal survival under low osmolarity conditions. However, we agree with the reviewer that it is formally possible that the plasmid-directed expression of rpoS in Ty2 could compensate for some other factor that makes this strain sensitive to low osmolarity. We have mentioned this possibility on p. 15, lines 451-454 of the revised manuscript in the context of the new findings shown in Fig. 7. 

 The reviewer also mentions that “it would be helpful to introduce the cloned rpoS into other S. Typhi mutant, low osmolarity sensitive strains” and that “it would be informative to test the S. Typhi strains/isolates with WT rpoS genes for low osmolarity resistance”.

 As mentioned above, we have tested the S. Typhi strain χ3744, which has a wild-type rpoS gene, and have shown that it is resistant to incubation under low osmolarity conditions (Fig. 7 of the revised manuscript). While we agree with the reviewer that it would also be helpful to express the cloned rpoS gene in other S. Typhi strains with mutant rpoS, this experiment is not easily done since we would first have to identify the mutant strains among multiple candidates, a process that would probably require sequencing the relevant genomic region. Moreover, the outcome of this effort would serve mainly to confirm our findings with Ty2. For these reasons, we respectfully submit that our results with rpoS expression in Ty2, together with the new findings generated using the rpoS wild-type, rpoS inactivated and rpoS inducible strains (Fig. 7), already provide strong evidence that rpoS plays an important role in resistance to low osmolarity. Thus, we feel that expression of rpoS in additional rpoS mutant strains is not essential to the scope of the present work.

---

## [Decision Letter · Decision Letter 1]

6 Dec 2022

The rpoS gene confers resistance to low osmolarity conditions in Salmonella enterica serovar Typhi

PONE-D-22-26135R1

Dear Dr. Cherayil,

We’re pleased to inform you that your manuscript has been judged scientifically suitable for publication and will be formally accepted for publication once it meets all outstanding technical requirements.

Kind regards,

Olivia Steele-Mortimer, Ph.D.

Academic Editor

PLOS ONE

Additional Editor Comments (optional):

Reviewers' comments:

Reviewer's Responses to Questions

**Comments to the Author**

1. If the authors have adequately addressed your comments raised in a previous round of review and you feel that this manuscript is now acceptable for publication, you may indicate that here to bypass the “Comments to the Author” section, enter your conflict of interest statement in the “Confidential to Editor” section, and submit your "Accept" recommendation.

Reviewer #1: All comments have been addressed

Reviewer #2: All comments have been addressed

2. Is the manuscript technically sound, and do the data support the conclusions?

Reviewer #1: Yes

Reviewer #2: Yes

3. Has the statistical analysis been performed appropriately and rigorously? 

Reviewer #1: Yes

Reviewer #2: Yes

4. Have the authors made all data underlying the findings in their manuscript fully available?

Reviewer #1: Yes

Reviewer #2: Yes

5. Is the manuscript presented in an intelligible fashion and written in standard English?

Reviewer #1: Yes

Reviewer #2: Yes

6. Review Comments to the Author

Reviewer #1: (No Response)

Reviewer #2: (No Response)

7. PLOS authors have the option to publish the peer review history of their article (what does this mean?). If published, this will include your full peer review and any attached files.

Reviewer #1: No

Reviewer #2: No

---

## [Editor Report · Acceptance letter]

8 Dec 2022

PONE-D-22-26135R1 

The rpoS gene confers resistance to low osmolarity conditions in Salmonella enterica serovar Typhi 

Dear Dr. Cherayil:

I'm pleased to inform you that your manuscript has been deemed suitable for publication in PLOS ONE. Congratulations! Your manuscript is now with our production department. 

Kind regards, 

on behalf of

Dr. Olivia Steele-Mortimer 

Academic Editor

PLOS ONE